# Design, Optimization, Manufacture and Characterization of Milbemycin Oxime Nanoemulsions

**DOI:** 10.3390/pharmaceutics17030289

**Published:** 2025-02-22

**Authors:** Ze-En Li, Yang-Guang Jin, Shao-Zu Hu, Yue Liu, Ming-Hui Duan, Shi-Hao Li, Long-Ji Sun, Fan Yang, Fang Yang

**Affiliations:** College of Animal Science and Technology, Henan University of Science and Technology, Luoyang 471023, China; lze1973535073@126.com (Z.-E.L.); jinyangguang1999@126.com (Y.-G.J.); 15038541381@163.com (S.-Z.H.); lyue0523@126.com (Y.L.); duanminghui0802@126.com (M.-H.D.); lishihao0610@126.com (S.-H.L.); s17538591074@126.com (L.-J.S.)

**Keywords:** nanoemulsions, milbemycin oxime, manufacture, phase inversion composition, in vitro release

## Abstract

**Background:** Despite the rapid development of nanoemulsions in recent years, no method has been established for the preparation of milbemycin oxime nanoemulsions. Milbemycin oxime is a widely used macrolide antibiotic in veterinary medicine, particularly for treating parasitic infections in animals such as dogs. However, its poor solubility in water limits its bioavailability and therapeutic efficacy. Developing a nanoemulsion formulation can enhance its solubility, stability, and bioavailability, offering a more effective treatment option. **Methods:** In this experiment, oil-in-water (O/W) milbemycin oxime nanoemulsions were successfully prepared by the phase inversion composition (PIC) method using ethyl butyrate as the oil phase, Tween-80 as the surfactant, and anhydrous ethanol as the co-surfactant. The region of O/W nanoemulsions was identified by constructing a pseudo-ternary phase diagram and, based on this, was screened by determining the droplet size, polydispersity coefficient, and zeta potential of each preparation. **Results and Conclusions:** The finalized formulation had a 2:1 ratio of surfactant to co-surfactant and a 7:3 ratio of mixed surfactant to oil, and its droplet size, polydispersity index (PDI), and zeta potential were 12.140 ± 0.128 nm, 0.155 ± 0.015, and −4.947 ± 0.768 mV, respectively. Transmission electron microscopy confirmed the spherical uniform distribution of droplets, and the nanoemulsions passed thermodynamic stability tests. The in vitro release of milbemycin oxime nanoemulsions followed first-order kinetic equations. In conclusion, nanoemulsions are an interesting option for the delivery of poorly water-soluble molecules such as milbemycin oxime.

## 1. Introduction

Milbemycin oxime, a macrolide antibiotic derived from the actinomycete *Streptomyces hygroscopicus*, consists of 80% A4 and 20% A3 derivatives of 5-didehydromilbemycin [1,2]. This compound demonstrates high efficacy against both internal and external parasites in dogs and cats, including hookworms, roundworms, and whipworms and external pests such as Trichinella, mange mites, lice, and fleas [3,4,5]. Milbemycin oxime exerts its anthelmintic activity by binding to high-affinity sites on target cells, increasing the permeability of cell membranes to chloride ions (Cl^−^). This action triggers the release of gamma-aminobutyric acid (GABA) in helminth neurons and arthropod myocytes, leading to the opening of glutamate-gated Cl^−^ channels and an increase in neuronal membrane permeability. The disruption of nerve signal transmission results in parasite paralysis and death [6].

The safety profile of milbemycin oxime has been well documented, with studies in over 75 dog breeds, including collies, pregnant dogs, and puppies older than two weeks, confirming its broad safety margin [7]. In acute toxicity tests, the median lethal dose (LD_50_) in adult dogs exceeded 200 mg/kg body weight (BW) [7]. Moreover, a study by Sankyo Co., Ltd. in Japan demonstrated that even in the most sensitive breed, the Collie, ten times the recommended dose (2.5 mg/kg BW daily) over ten days caused no significant alterations in liver function markers such as aspartate aminotransferase (AST) and alanine aminotransferase (ALT). These findings underscore the drug’s wide safety margin and reinforce its reliability as a therapeutic option for dogs [8].

Milbemycin oxime offers substantial economic and social benefits, owing to its high safety profile and broad-spectrum anthelmintic activity. However, it is currently available only in regular tablets, oral chewable tablets, and a 0.1% ear solution for ear mites [9]. In China, the tablet form is the sole available option, yet it suffers from poor intestinal absorption, leading to suboptimal therapeutic outcomes [10]. This limitation highlights the pressing need for new dosage forms to provide more effective deworming alternatives for veterinarians and pet owners.

In recent years, nanoemulsions have emerged as a promising technology in drug formulation, particularly for enhancing the bioavailability of poorly soluble drugs [11,12]. Nanoemulsion-based delivery systems are characterized by droplet sizes ranging from 1 to 100 nm [13], and their thermodynamic stability, isotropic nature, and transparency make them ideal candidates for drug delivery applications [14,15]. Nanoemulsions have been shown to improve solubility, facilitate penetration through cellular gaps and epithelial barriers, and enhance drug stability, which in turn enables targeted delivery and controlled release [16,17,18]. Moreover, they reduce the efflux of drugs by P-glycoprotein (P-gp) and increase drug affinity for intestinal membranes, which significantly improves the oral bioavailability of poorly soluble drugs [19,20].

Several studies have demonstrated the potential of nanoemulsion systems to enhance the oral absorption of drugs with poor solubility. For instance, oral nanoemulsions of florfenicol exhibited a peak concentration (C_max_) 3.42 times higher than the commercial formulation, with a relative bioavailability of 134.5% in pigs [21]. Similarly, nanoemulsions of bicyclol increased the area under the curve (AUC) by 7.7 times and C_max_ by 7.2 times compared with the suspension form, resulting in a 7.7-fold increase in bioavailability in rats [22]. For fenofibrate, oral nanoemulsions showed 1.63 times the bioavailability of commercial capsules in dogs [23]. Furthermore, nanoemulsions have been shown to enhance the affinity of puerarin for intestinal membranes, further supporting their ability to improve drug absorption [20]. These findings highlight the significant potential of nanoemulsion systems in improving the oral bioavailability of poorly soluble drugs.

Despite these promising results, a nanoemulsion formulation for milbemycin oxime has not yet been developed. Given the compound’s therapeutic potential and the challenges associated with its poor solubility, the main focus of the current research is to develop an oil-in-water (O/W) nanoemulsion formulation for milbemycin oxime. This study aims to optimize the formulation, evaluate its physicochemical properties, and assess the in vitro release kinetics, with the goal of improving the bioavailability and therapeutic efficacy of milbemycin oxime in veterinary applications.

## 2. Materials and Methods

### 2.1. Drugs and Chemicals

The analytical standard of milbemycin oxime (Lot No. NO9GS166952; 99% purity) was obtained from Shanghai Yuanye Bio-Technology Co., Ltd. (Shanghai, China), while the raw material (Lot No. 202311-1; 99% purity) was sourced from Livzon Group Fuzhou Fuxing Pharmaceutical Co., Ltd. (Fuzhou, Fujian, China). Isopropyl myristate (IPM), soybean oil, ethyl oleate, butyl acetate, ethyl butyrate, clove oil, diethyl malonate, oleate, Tween-80, RH-40, Tween-20, anhydrous ethanol, glycerol, polyethylene glycol-400, isopropanol, 1,2-propylene glycol, ammonium acetate, methanol (chromatographic grade), and acetonitrile (chromatographic grade) were acquired from Shanghai Macklin Biochemical Technology Co., Ltd. (Shanghai, China). Water was purified using a Milli-Q ultrapure water system (Millipore Corp., Shanghai, China). All other reagents were sourced from commercial suppliers.

### 2.2. HPLC Analysis

Milbemycin oxime was quantitatively analyzed using a Waters e2695 high-performance liquid chromatography (HPLC) system equipped with a 2489 UV detector and Empower software (Version 8.1; Waters Corporation, Milford, MA, USA). Chromatographic separation was achieved using a Hypersil BDS C18 column (4.6 mm × 250 mm, 5 μm; Elite Analytical Instruments Co., Ltd., Dalian, China). The mobile phase consisted of 14% 0.5 mmol/L ammonium acetate buffer and 86% acetonitrile, with a flow rate of 1 mL/min. The detection wavelength was set at 249 nm, and the column temperature was maintained at 25 °C. A 20 μL injection volume was used for all analyses. The method demonstrated linearity over the concentration range of 0.1–200 μg/mL (R^2^ = 0.999) and precision, with a % RSD < 1.35% for all samples tested. The limit of quantification (LOQ) was 0.05 μg/mL, and the limit of detection (LOD) was 0.025 μg/mL.

### 2.3. Solubility of Milbemycin Oxime in Each Excipient

The solubility of milbemycin oxime was determined by adding an excess of the drug to 1 mL of various oils (IPM, soybean oil, ethyl oleate, butyl acetate, ethyl butyrate, clove oil, diethyl malonate, oleate), surfactants (Tween-80, RH-40, Tween-20), and co-surfactants (anhydrous ethanol, glycerol, polyethylene glycol-400, isopropanol, 1,2-propylene glycol) and allowing the mixture to stir magnetically for 48 h at 25 ± 1.0 °C [24].

The equilibrated samples were centrifuged at 10,000 rpm for 15 min to separate the undissolved drug. The supernatant was then filtered through a 0.45 μm filter membrane. The concentration of milbemycin oxime in each excipient was determined at 249 nm using HPLC–UV, following the method described in Section 2.2.

### 2.4. Screening of Oil Phases

In Section 2.3, we assessed the solubility of milbemycin oxime in various excipients. Since milbemycin oxime is a water-insoluble substance, enhancing its solubility in nanoemulsions is crucial for improving therapeutic efficacy and potentially reducing the required dosage. Based on the results, Tween-80 (surfactant) and anhydrous ethanol (co-surfactant) were selected for their superior solubility. The oil phase also plays a key role in nanoemulsion formation, as effective interaction between the oil molecules and the interfacial film is essential. Therefore, different oil phases were used to prepare nanoemulsions with the selected surfactant mixture (Tween-80 and anhydrous ethanol), as described in Section 2.5.

### 2.5. Construction of Pseudo-Ternary Phase Diagram

After selecting anhydrous ethanol as the co-surfactant and Tween-80 as the surfactant, different oil phases were mixed with the surfactant mixture (S_mix_). Then, a pseudo-ternary phase diagram was constructed via water titration to identify the nanoemulsion zone and optimize the formulation, with ethyl butyrate chosen as the best oil phase. Surfactant-to-co-surfactant ratios (Km) of 1:1, 2:1, and 3:1 were tested at room temperature. The surfactant mixture was combined with ethyl butyrate at ratios of 9.5:0.5, 9:1, 8.75:1.25, 8.5:1.5, 8:2, 7.5:2.5, and 7:3, as ratios below 6:4 could not form nanoemulsions.

The mixture of S_mix_ and ethyl butyrate was thoroughly combined, and milbemycin oxime was added. Once the milbemycin oxime was fully dissolved, distilled water was added drop by drop. During this process, the water should be added slowly while stirring continuously, ensuring that the oil and water phases are fully mixed. The change of the state of the system was observed throughout the process. As the water phase was added, viscosity increased, and greater resistance to mixing was encountered. Upon reaching a certain amount of water, the system suddenly became more fluid. The system transitioned from turbid to clear or vice versa, and the critical point of this change, in terms of water addition, was recorded. The mass percentage of each component in the system was then calculated, and the Tyndall effect was observed at this point.

Pseudo-ternary phase diagrams were drawn using Origin software (version 2018; OriginLab) with the three vertices of the phase diagrams of the surfactant mixture (S_mix_), the oil phase (Oil), and the water phase (Water) as the three vertices. Each edge of the diagrams indicated the proportionality of the corresponding two components, and any one point indicated the mass percentage of each component. The size of the nanoemulsion region was used to determine the optimal preparation for milbemycin oxime nanoemulsions. In this study, only the O/W nanoemulsion region was plotted in the pseudo-ternary phase diagram.

### 2.6. Optimization of Preparations and Droplet Size

By plotting the pseudo-ternary phase diagram, we identified a Km value of 2:1, which corresponded to the largest nanoemulsion region. To optimize the formulation, we then measured the droplet size, polydispersity index (PDI), and zeta potential of each formulation.

The mean droplet size and PDI of the nanoemulsions were measured using dynamic light scattering (DLS) with a Malvern 3000 photon correlation spectrometer (Malvern Instruments, Malvern, UK), equipped with a 488 nm argon laser. Measurements were taken at a scattering angle of 90° and a constant temperature of 25 °C. The cumulants method was used to analyze the DLS data and determine the z-average diameter and PDI. For the zeta potential measurements, samples were injected into a DT51070 folded capillary cell (Malvern Instruments). To minimize multiple scattering, samples were diluted 1:10 in double-distilled water before measurement.

Generally, smaller droplet sizes in nanoemulsion systems correlate with higher bioavailability [25], while a lower PDI indicates a more homogeneous and stable sample [26]. Additionally, emulsions with a higher zeta potential demonstrate greater physical stability [27].

### 2.7. Properties of Milbemycin Oxime Nanoemulsion

The milbemycin oxime nanoemulsions were placed in a transparent container for observation of color, turbidity, and transparency. To assess their dilution capacity, the nanoemulsions were mixed with water and then centrifuged at 10,000 rpm for 15 min to check for layering. Additionally, the Tyndall effect was examined by observing the nanoemulsions under converging light, as colloidal emulsions with droplet sizes of 1 to 100 nm produce a visible band of light.

For nanoemulsion identification, we employed the staining method, which is effective and straightforward. Methylene blue and Sudan red were selected as reagents. Methylene blue, being a water-soluble substance, rapidly diffuses in O/W emulsions, creating observable phenomena [28], while it is not visible in W/O emulsions. To conduct the staining, we added the appropriate amount of methylene blue or Sudan red solution to the nanoemulsions in EP tubes and recorded the observations.

### 2.8. Morphological Examination

Transmission electron microscopy (TEM) was utilized to analyze the shape and surface morphology of droplets in both nanoemulsions and aqueous dispersions. A JEM-F200 TEM (JEOL Co., Tokyo, Japan) operating at 200 kV was used for the analysis. Samples were deposited on a carbon-coated copper grid, left to stand for 10 min at room temperature, stained with 2% uranyl acetate for 5 min, dried, and then examined under the TEM. Prior to measurement, samples were diluted 1:10 in double-distilled water. The droplet sizes were obtained by analyzing the TEM images using Nano Measurer 1.2 software (Department of Chemistry, Fudan University, Shanghai, China).

### 2.9. Stability Tests

Nanoemulsions are thermodynamically stable systems formed at specific concentrations of oil, surfactant, co-surfactant, and water, making them resistant to phase separation, wrinkling, or cracking. To investigate the thermodynamic stability of the screened formulations, we employed centrifugation, heating–cooling cycles, and freeze–thaw cycles [29].

Centrifugation Studies: Samples were centrifuged at 10,000 rpm for 15 min to assess stability under high speeds.

Heating–Cooling Cycles: Samples underwent six heating–cooling cycles, alternating between 4 °C and 45 °C over 48 h, to evaluate thermal stability.

Freeze–Thaw Cycles: Formulations were subjected to three freeze–thaw cycles between −21 °C and +25 °C for 48 h, with observations recorded for any signs of instability, such as precipitation or phase separation.

In addition to the thermodynamic stability testing, we also performed long-term stability testing on the milbemycin nanoemulsion. The stability study was conducted at 25 ± 2 °C and a relative humidity of 60% ± 10%. Samples were taken at 0, 1, 3, 6, and 9 months to assess the properties of the nanoemulsion and to quantify the content of milbemycin oxime using the method developed in Section 2.2.

### 2.10. Determination of Encapsulation Efficiency and Drug Loading

A 1 mL sample of milbemycin oxime nanoemulsion was precisely measured and added to an ultrafiltration tube (Da: 10,000) and then centrifuged at 3500 rpm for 15 min. The lower filtrate was collected, diluted with acetonitrile, and analyzed by HPLC to record the peak area and calculate the free drug concentration (C_f_). Another 1 mL sample of milbemycin oxime nanoemulsion was mixed with acetonitrile and sonicated for 5 min to break the emulsion. The mixture was centrifuged at 12,000 rpm for 10 min, diluted with acetonitrile, filtered through a 0.45 µm membrane, and analyzed by HPLC. The peak area was recorded to calculate the total drug concentration (C_t_). Finally, encapsulation efficiency (EE%) and drug loading (DL%) were then determined using Equations (1) and (2) [30]:(1)$$\mathrm{EE}\%={(C}_{t}\;-\;C_{f})/C_{t}\text{×}100\%$$(2)$$\mathrm{DL}\%=(C_{t}-\;C_{f})/M\text{×}100\%$$
where C_t_ is the total drug concentration in the formulation; C_f_ is the free drug concentration; and M is the total amount of milbemycin oxime in the nanoemulsion.

### 2.11. In Vitro Release Studies

In vitro release studies were performed in simulated intestinal fluid, composed of 0.3% bile salts, 0.9% NaCl, 1% trypsin, and distilled water [31]. A regenerated cellulose dialysis membrane with a molecular weight cutoff of 10,000 Da was used for the drug release studies. Precisely 1 g of the test solution was placed into a bubble-free dialysis bag, which was securely sealed using clips. The dialysis bag was immersed in 100 mL of enteric fluid containing pancreatic lipase (4.8 mg/mL). To simulate the intestinal environment, the solution pH was adjusted to 7 using 1 M NaOH. The experiment was conducted at 37 °C with continuous stirring at 100 rpm for 48 h to mimic digestive system movement.

Samples (1 mL) were withdrawn at intervals of 15, 30, and 45 min and then at 1, 2, 3, 4, 6, 8, 12, 24, 36, and 48 h and replaced with an equal volume of fresh simulated digestive fluid. The withdrawn samples were analyzed using HPLC to determine the concentration of released milbemycin oxime. The cumulative drug release percentage was calculated using Equation (3) [32]:(3)$$\begin{array}{c} Cumulative\;drug\;release\left(\%\right)=\frac{V_{c}{{\text{∑}}_{1}}^{n-1}C_{i}+V_{t}C_{n}}{C_{0}V_{0}}\text{×}100\% \end{array}$$
where V_c_ is the volume of medium removed at each time point (1 mL); V_t_ is the total volume of the system (100 mL); C_i_ is the concentration of milbemycin oxime in the medium; C_0_ is the concentration of milbemycin oxime in the nanoemulsion formulation (10 mg/mL); and V_0_ is the volume of the nanoemulsion formulation (1 mL).

To assess the release kinetics of milbemycin oxime from the nanoemulsion formulation, the release data were fitted to zero-order kinetics, first-order kinetics, the Higuchi model, and the Korsmeyer–Peppas model.

To study the release kinetics of milbemycin oxime in the formulation, mathematical models such as zero-order kinetics (Equation (4)) and first-order kinetics (Equation (5)) were applied:(4)$$\begin{array}{c} \frac{M_{t}}{M_{\text{∞}}}=k_{0}t \end{array}$$(5)$$\begin{array}{c} M_{t}\;=M_{\text{∞}}(1-e^{-k_{1}t}) \end{array}$$
where M_t_ is the cumulative release at time t; M_∞_ is the cumulative release at ∞; M_t_/M_∞_ is the cumulative percentage release at time t; and k_0_ and k_1_ are the rate constants for the zero-order kinetics and first-order kinetics, respectively.

The Higuchi model is commonly applied in controlled release systems, particularly for matrix-based formulations. This model helps determine whether drug release follows Fickian diffusion, as described in Equation (6). The Korsmeyer–Peppas model is used to explain drug release from hydrophilic polymer matrices by plotting the logarithmic cumulative percentage of drug released over time, as shown in Equation (7).(6)$$\begin{array}{c} \frac{M_{t}}{M_{\text{∞}}}=k_{H}t^{\frac{1}{2}} \end{array}$$(7)$$\begin{array}{c} \frac{M_{t}}{M_{\text{∞}}}=k_{\mathrm{KP}}t^{n} \end{array}$$

Here, n is the diffusion index; and k_H_ and k_KP_ are the drug release rate constants of the Higuchi model and the Korsmeyer–Peppas model, respectively.

The kinetic parameters were estimated from the time (hours) versus the cumulative release (%) curve. To identify the best-fit model, we compared the correlation coefficient (R^2^) values obtained through linear or nonlinear regression analysis of various mathematical models. The model with an R^2^ value closest to 1.0 was considered the most suitable for describing the release kinetics.

## 3. Results and Discussion

### 3.1. Solubility Analysis

The solubility analysis results shown in Table 1 indicate that milbemycin oxime has the highest solubility in anhydrous ethanol, Tween-80, and ethyl butyrate. Surfactants, as an important component of nanoemulsion systems, provide stability to the nanoemulsion system mainly by reducing the interfacial tension [33]. The smaller the size of the surfactant the easier it is to reduce the interfacial tension of the emulsion system to form nanoemulsions. Tween-80 is also widely chosen for its ability to produce smaller droplets and has the best surfactant molecular geometry compared with other Tweens [34]. In the case of Tween-20, phase separation due to its hydrophilic structure is a challenge to the stability of nanoemulsions [35]. Due to the significant variation in solubility between different surfactants and co-surfactants, we selected Tween-80 as the surfactant and anhydrous ethanol as the co-surfactant to enhance the loading capacity of the milbemycin oxime nanoemulsions. To identify a suitable oil phase, we then evaluated the ability of various oil phases to form nanoemulsions with Tween-80 and anhydrous ethanol.

### 3.2. Results of Oil Phase Screening

Various oil phases were mixed with Tween-80 as the surfactant and anhydrous ethanol as the co-surfactant. Distilled water was then added dropwise to the mixtures to construct pseudo-ternary phase diagrams and prepare the nanoemulsions. The screening results are presented in Table 2. As indicated, clarified and transparent nanoemulsions were formed with ethyl butyrate within a specific range, while diethyl malonate also produced nanoemulsions, though they lacked sufficient stability. Based on these findings, ethyl butyrate was selected as the oil phase for the milbemycin oxime nanoemulsions.

### 3.3. Pseudo-Ternary Phase Diagram Analysis

Currently, two main methods are employed to prepare nanoemulsions: high-energy and low-energy emulsification techniques. Low-energy methods include the spontaneous phase inversion temperature (PIT) and phase inversion composition (PIC) techniques [36]. The PIT method involves temperature changes at a constant composition, whereas the PIC method focuses on composition changes at a constant temperature. In this experiment, the PIC method was applied to construct pseudo-ternary phase diagrams. The principle of the PIC method is to continuously add the water phase to the oil phase. At the beginning, due to the excess amount of oil, W/O nanoemulsion is formed. As the proportion of the water phase increases, the curvature of the surfactant is changed, and the water droplets gradually polymerize together. At the transition point of the emulsion phase, the surfactant forms a layer structure, and the surface tension is at minimum, which is conducive to the formation of fine dispersed emulsion droplets. After the phase transition point of emulsion, with the further addition of water, O/W nanoemulsion is formed, which is the emulsion droplet formation process in the PIC method [37]. Compared with the high-energy emulsification method, no high shear equipment is required, the cost is low, the energy consumption is low, and the emulsion stability is good because it does not need a high temperature environment for heat-sensitive substances.

The results of the water addition titration method are presented in Table 3. For convenience, the mass percentages of the components were normalized to plot the pseudo-ternary phase diagrams. As shown in Table 3, it was difficult to form nanoemulsions when the Km value was 3:1 and the S_mix_:Oil ratio was 7:3. The pseudo-ternary phase diagrams, depicted in Figure 1, reveal that the nanoemulsion area when Km = 1:1 is similar to that formed at Km = 2:1 but significantly larger than the area at Km = 3:1. Consequently, we measured the droplet size, polydispersity index (PDI), and zeta potential of the nanoemulsions prepared at Km values of 1:1 and 2:1.

### 3.4. Optimize Preparation

The droplet size, PDI, and zeta potential of each formulation are summarized in Table 4. When a Km value of 2:1 and S_mix_:Oil ratios of 7.5:2.5 or 7:3 were used, the resulting nanoemulsions exhibited emulsion breakage upon dilution within 24 h. Consequently, we focused on screening the optimal formulation among those with a Km value of 2:1. These nanoemulsions were evaluated in terms of the droplet size, PDI, and zeta potential.

The bioavailability of encapsulated bioactive compounds tends to increase as droplet size decreases within the emulsion system [36]. In nanoemulsions, the PDI describes the width of the droplet size distribution; a smaller PDI indicates a more concentrated droplet size distribution [38]. Zeta potential serves as an indicator of the stability of a colloidal system: when droplets possess a high negative or positive charge, resulting in a very high zeta potential, they repel each other, thereby enhancing the stability of the entire system. Consequently, we selected the formulation with Km = 2:1 and S_mix_:Oil = 7:3 as the optimal milbemycin oxime nanoemulsion formulation. Figure 2 presents the particle size detection plot of this formulation, while Figure 3 shows the zeta potential detection plot of the modified formulation. The optimized formulation exhibited a droplet size of 12.140 ± 0.128 nm, a PDI of 0.155 ± 0.015, and a zeta potential of −4.947 ± 0.768 mV.

In general, zeta potential values greater than +30 mV or less than −30 mV indicate high stability [39]. Given the lower zeta potential of our nanoemulsions, we evaluated the stability of the optimal formulation over a period of nine months at 25 °C ± 2 °C and a relative humidity of 60% ± 10%. Also as a part of the long-term stability tests, we monitored changes in the particle size, PDI, zeta potential, and drug content. Detailed results and discussion are presented in Section 3.7.

### 3.5. Studies on the Properties of Milbemycin Oxime Nanoemulsions

The prepared milbemycin oxime nanoemulsions appeared as a light yellow, clarified, transparent emulsion (Appendix A). Upon dilution with water, these nanoemulsions exhibited the property of infinite dilution. Centrifugation at 10,000 rpm for 15 min showed no delamination. When irradiated with infrared light, a clear light path was observed, demonstrating the Tyndall effect (Appendix A). In the staining experiment, the addition of methylene blue dye resulted in rapid diffusion of the blue color throughout the nanoemulsions, while the Sudan red dye did not spread (Appendix A). This indicates that the milbemycin oxime nanoemulsions produced in this experiment are of the O/W type.

### 3.6. TEM Analysis

Electron microscopy is an effective tool for characterizing nanoemulsions, as it can visualize nanostructures that conventional microscopy techniques cannot detect [40]. The milbemycin oxime nanoemulsion formulations were analyzed using TEM to assess droplet morphology and distribution, as shown in Figure 4. The droplets were spherical or elliptical in shape, and the TEM images were analyzed using Nano Measurer 1.2 software to obtain a droplet size of 16.069 ± 2.951 nm, which is larger than the 12.140 ± 0.128 nm obtained by the DLS method, and these errors may be due to the fact that the TEM measures the droplet sizes in the dry state. The two principles are different and the results are naturally different. Importantly, the droplets are non-aggregated, indicating high stability of the nanoemulsion and suggesting that it does not undergo Ostwald maturation due to spherical collapse [41].

### 3.7. Stability Studies

The thermodynamic stability study assessed the stability of the screened formulations. The nanoemulsions exhibited excellent resilience, with no precipitation or phase separation observed during centrifugation studies, heating–cooling cycles, or freeze–thaw cycles.

The change results of the particle size, PDI, zeta potential, and drug content in the long-term stability test are shown in Table 5, where the drug content at day 0 is 100%. The results show that the droplet size decreases over time, the PDI also decreases, and the zeta potential shows a gradual increase, which is an indication that the nanoemulsion is becoming more stable. Although the drug content decreased gradually, it remained above 90% for nine months, indicating that the milbemycin oxime nanoemulsion remained stable and potent for at least nine months.

### 3.8. Encapsulation Efficiency and Drug Loading Studies

Three milbemycin oxime nanoemulsions were prepared in parallel using the optimal formulation, followed by ultrafiltration centrifugation and HPLC analysis to measure EE% and DL%, as shown in Table 6. The results demonstrated complete encapsulation of milbemycin oxime within the oil phase of the nanoemulsion system, indicating high formulation efficiency. This effectiveness may be attributed to the lipophilicity of milbemycin oxime, which helps retain the drug in the oil core. The optimal formulation achieved an average EE% of 99.153 ± 0.482% and an average DL% of 1.001 ± 0.002%.

### 3.9. In Vitro Drug Release and Release Kinetics

To improve the therapeutic efficacy of milbemycin oxime and optimize drug delivery, we have used O/W nanoemulsions to encapsulate the drug in oil. It is essential to monitor the release of the drug from these nanoemulsions. In vitro drug release and release kinetics were employed to evaluate the ability of the nanoemulsion formulations to release milbemycin oxime by measuring the amount released over time. The cumulative release of milbemycin oxime was assessed at each time point using dynamic dialysis, resulting in a time (h) versus cumulative release (%) curve. The mean ± SD release profile of the milbemycin oxime nanoemulsions in simulated intestinal fluid is depicted in Figure 5. Kinetic models—Zero-order, First-order, Higuchi, and Korsmeyer–Peppas—were fitted to the in vitro release profile using Origin 2018 software, and the corresponding R^2^ values are detailed in Table 7.

However, using the mean release curve for model fitting often results in unsatisfactory R^2^ values. For instance, when applying a first-order kinetic model, the R^2^ value from the mean release curve was 0.97. In contrast, fitting the model using three separate release curves resulted in an R^2^ value of 0.99. Consequently, we opted to fit each release curve individually, and the results are presented in Appendix A.

Comparing the R^2^ values of the Zero-order, First-order, Higuchi, and Korsmeyer–Peppas models showed that the first-order kinetic model had the highest R^2^ value of 0.99, indicating that it was the best fit for the release profile.

## 4. Conclusions

We present a method for preparing a new dosage form of milbemycin oxime nanoemulsion and optimizing its formulation through a series of experiments. By using Tween-80 as a surfactant, anhydrous ethanol as a co-surfactant, and ethyl butyrate as the oil phase, we successfully obtained milbemycin oxime nanoemulsions characterized by a clarified transparent appearance and spherical uniformly distributed droplets, as verified by the PIC method, TEM, and DLS. The optimal formulation was identified with a mass ratio of Tween-80 to anhydrous ethanol of 2:1 and a ratio of S_mix_ to Oil of 7:3. This formulation exhibited a droplet size of 12.140 ± 0.128 nm, a PDI of 0.155 ± 0.015, and a zeta potential of −4.947 ± 0.768 mV, demonstrating stability for at least five months. In addition, it achieved an average encapsulation efficiency (EE%) of 99.153 ± 0.482% and a drug loading (DL%) of 1.001 ± 0.002%. The in vitro release of the formulation adhered to a first-order kinetic model, with an R^2^ value of 0.99, indicating a highly fitting cumulative drug release profile. The formulation of the insoluble drug milbemycin oxime into nanoemulsions significantly enhances its solubility and is expected to exploit the advantages of nanoemulsion formulations, providing a promising new option for clinical applications.

## Figures and Tables

**Figure 1 pharmaceutics-17-00289-f001:**
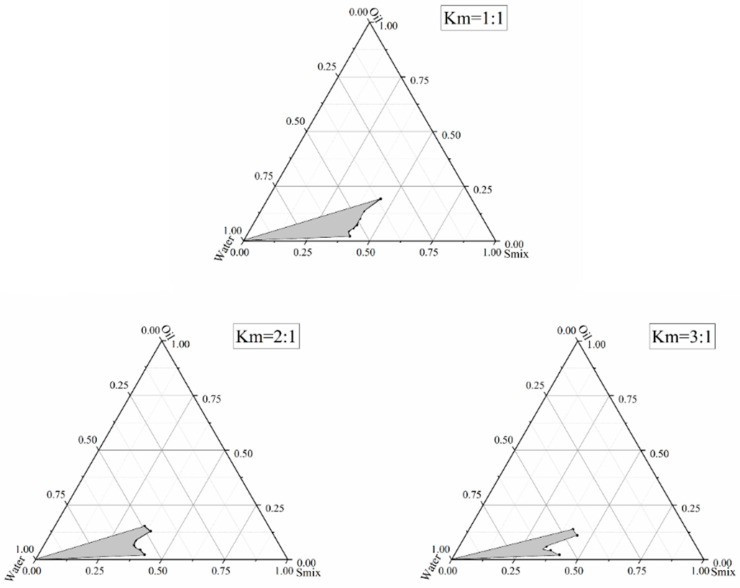
The pseudo-ternary phase diagram under different conditions. The shaded area is the O/W nanoemulsion region.

**Figure 2 pharmaceutics-17-00289-f002:**
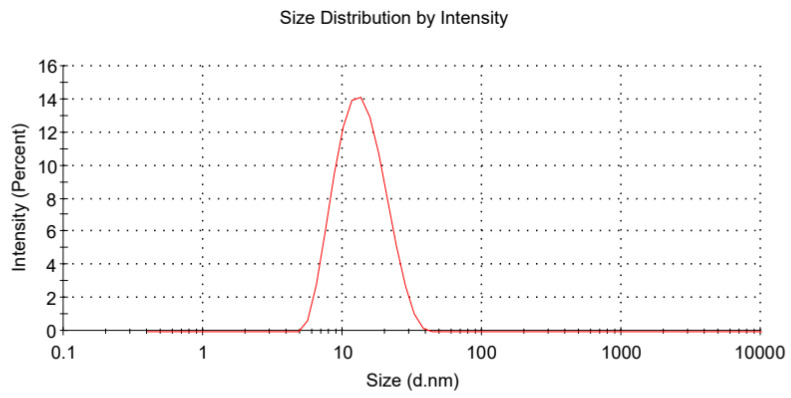
Size distribution of milbemycin oxime nanoemulsions measured by dynamic light scattering (DLS) (one measurement).

**Figure 3 pharmaceutics-17-00289-f003:**
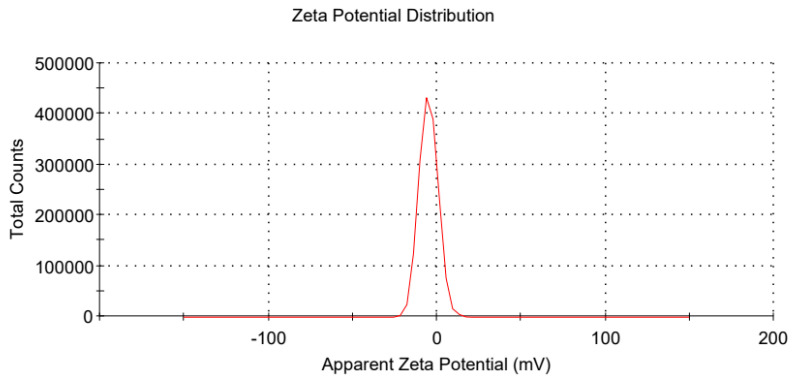
Zeta potential images of milbemycin oxime nanoemulsions (one measurement).

**Figure 4 pharmaceutics-17-00289-f004:**
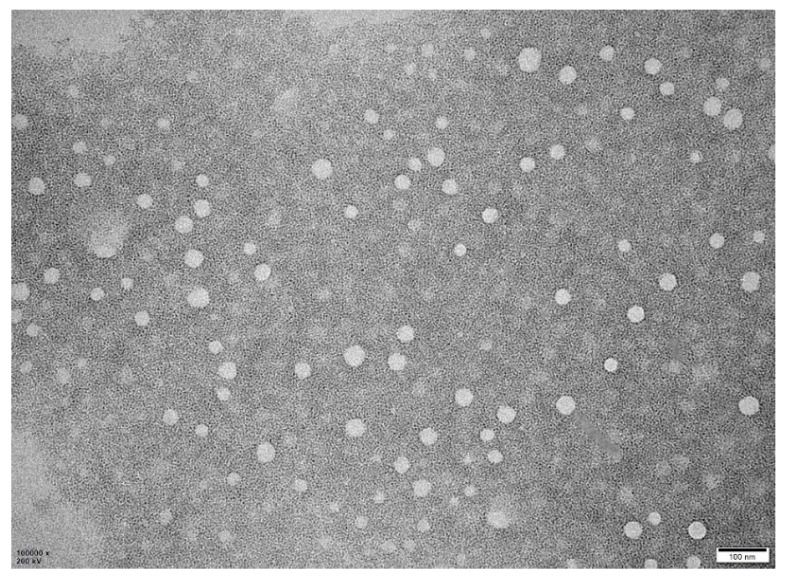
TEM of milbemycin oxime nanoemulsions (×100,000 magnification).

**Figure 5 pharmaceutics-17-00289-f005:**
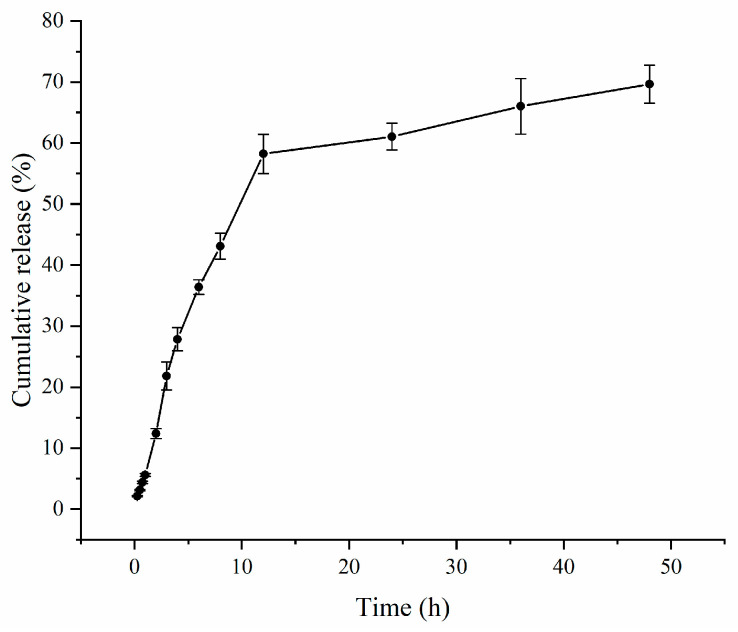
Mean ± SD release profiles of milbemycin oxime nanoemulsions in simulated intestinal fluids (n = 3).

**Table 1 pharmaceutics-17-00289-t001:** Mean ± SD solubility of milbemycin oxime in different excipients (n = 3).

No.	Excipient	Function in Microemulsion	Solubility (mg/g)
1	Anhydrous ethanol	Co-surfactant	136.09 ± 1.33
2	Glycerol	Co-surfactant	0.30 ± 0.01
3	Polyethylene glycol-400	Co-surfactant	18.43 ± 0.26
4	Isopropanol	Co-surfactant	72.41 ± 2.02
5	1,2 Propylene glycol	Co-surfactant	41.67 ± 0.58
6	Tween-80	Surfactant	25.92 ± 0.30
7	RH-40	Surfactant	10.51 ± 0.20
8	Tween-20	Surfactant	3.48 ± 0.06
9	IPM	Oil	27.47 ± 0.12
10	Soya bean oil	Oil	9.05 ± 0.04
11	Ethyl oleate	Oil	21.62 ± 0.12
12	Butyl acetate	Oil	118.32 ± 1.37
13	Ethyl butyrate	Oil	132.43 ± 1.90
14	Clove oil	Oil	60.70 ± 2.08
15	Diethyl malonate	Oil	110.20 ± 1.95
16	Oleate	Oil	20.59 ± 0.59

**Table 2 pharmaceutics-17-00289-t002:** Results of screening for suitable oil phases in nanoemulsion preparation.

Oil Phases	Formation of Nanoemulsions
IPM	-
Soya bean oil	-
Ethyl oleate	-
Butyl acetate	-
Ethyl butyrate	+
Clove oil	-
Diethyl malonate	±
Oleate	-

Note: “+” means nanoemulsions can be made; “-” means nanoemulsions cannot be made; and “±” means nanoemulsions can be made but are unstable.

**Table 3 pharmaceutics-17-00289-t003:** The percentage of each component in different formulations when Km is 1:1, 2:1, and 3:1, respectively.

Km	S_mix_:Oil	Percentage		
		S_mix_	Oil	Water
1:1	9.5:0.5	0.4107	0.0216	0.5676
	9:1	0.3946	0.0438	0.5616
	8.75:1.25	0.4069	0.0581	0.5349
	8.5:1.5	0.4131	0.0729	0.5140
	8:2	0.4100	0.1025	0.4875
	7.5:2.5	0.4091	0.1364	0.4545
	7:3	0.4481	0.1920	0.3599
2:1	9.5:0.5	0.4207	0.0221	0.5571
	9:1	0.3935	0.0437	0.5627
	8.75:1.25	0.3691	0.0527	0.5782
	8.5:1.5	0.3581	0.0632	0.5787
	8:2	0.3530	0.0883	0.5587
	7.5:2.5	0.3910	0.1303	0.4787
	7:3	0.3562	0.1524	0.4914
3:1	9.5:0.5	0.4165	0.0219	0.5616
	9:1	0.3720	0.0413	0.5867
	8.75:1.25	0.3365	0.0481	0.6154
	8.5:1.5	0.3454	0.0610	0.5936
	8:2	0.4427	0.1107	0.4467
	7.5:2.5	0.4127	0.1375	0.4498
	7:3	NA	NA	NA

NA: no nanoemulsions could be formed; therefore, no data were available.

**Table 4 pharmaceutics-17-00289-t004:** Droplet size, PDI, and zeta potential detection for each formula (n = 3).

Km	S_mix_:Oil	Droplet Sizes (nm)	PDI	Zeta Potentials (mV)
1:1	9.5:0.5	11.483 ± 0.200	0.121 ± 0.027	−3.187 ± 0.012
	9:1	15.460 ± 0.300	0.247 ± 0.006	−3.483 ± 1.967
	8.75:1.25	14.313 ± 0.371	0.229 ± 0.003	−3.970 ± 0.707
	8.5:1.5	14.143 ± 0.363	0.232 ± 0.009	−2.890 ± 0.345
	8:2	15.575 ± 1.506	0.270 ± 0.045	−3.733 ± 0.947
	7.5:2.5	-	-	-
	7:3	-	-	-
2:1	9.5:0.5	15.907 ± 0.274	0.213 ± 0.017	−4.867 ± 1.060
	9:1	12.686 ± 0.669	0.220 ± 0.020	−3.950 ± 0.438
	8.75:1.25	12.163 ± 0.326	0.199 ± 0.018	−4.170 ± 0.564
	8.5:1.5	12.177 ± 0.292	0.182 ± 0.023	−4.253 ± 0.520
	8:2	17.410 ± 0.881	0.330 ± 0.016	−2.330 ± 0.339
	7.5:2.5	14.453 ± 0.506	0.273 ± 0.022	−3.893 ± 0.525
	7:3	12.140 ± 0.128	0.155 ± 0.015	−4.947 ± 0.768

Note: “-” means nanoemulsions break up after dilution within 24 h.

**Table 5 pharmaceutics-17-00289-t005:** Stability evaluation of milbemycin oxime nanoemulsions stored at 25 ± 2 °C and 60% ± 10% relative humidity for 9 months (n = 3).

Examined Items	0 Day	1 Month	3 Months	6 Months	9 Months
Droplet size (nm)	12.140 ± 0.128	11.973 ± 0.273	10.970 ± 0.072	10.880 ± 0.082	10.683 ± 0.004
PDI	0.155 ± 0.015	0.145 ± 0.023	0.0689 ± 0.007	0.059 ± 0.004	0.081 ± 0.005
Zeta potential (mV)	−4.947 ± 0.768	−5.090 ± 0.234	−5.767 ± 0.918	−6.700 ± 0.378	−8.060 ± 0.331
Content (%)	100	99.095 ± 0.056	97.292 ± 0.066	94.608 ± 0.042	91.931 ± 0.233

**Table 6 pharmaceutics-17-00289-t006:** Mean ± SD EE% and DL% of milbemycin nanoemulsions (n = 3).

Specimens	EE%	DL%
MBO-NE	99.153 ± 0.482	1.001 ± 0.002

Abbreviation: MBO-NE, milbemycin oxime nanoemulsion.

**Table 7 pharmaceutics-17-00289-t007:** In vitro release equation fitting of milbemycin oxime nanoemulsions (repeated three times).

Number	Models	Fitted Equation	R^2^
MBO-NE1	Zero-order	$\frac{M_{t}}{M_{\text{∞}}}\;$ = 14.64t + 1.35	0.71
MBO-NE2	Zero-order	$\frac{M_{t}}{M_{\text{∞}}}$ = 15.63t + 1.49	0.72
MBO-NE3	Zero-order	$\frac{M_{t}}{M_{\text{∞}}}\;$ = 16.55t + 1.46	0.70
MBO-NE1	First-order	$M_{t}$ = 63.94 ∗ (1 − e^−0.13t^)	0.99
MBO-NE2	First-order	$M_{t}$ = 69.85 ∗ (1 − e^−0.12t^)	0.99
MBO-NE3	First-order	$M_{t}$ = 69.61 ∗ (1 − e^−0.13t^)	0.99
MBO-NE1	Higuchi	$\frac{M_{t}}{M_{\text{∞}}}$ = 11.03t^1/2^ + 0.29	0.89
MBO-NE2	Higuchi	$\frac{M_{t}}{M_{\text{∞}}}$ = 12.06t^1/2^ − 0.004	0.90
MBO-NE3	Higuchi	$\frac{M_{t}}{M_{\text{∞}}}$ = 11.96t^1/2^ + 0.92	0.88
MBO-NE1	Korsmeyer–Peppas	$\frac{M_{t}}{M_{\text{∞}}}$ = 13.30t^0.44^	0.90
MBO-NE2	Korsmeyer–Peppas	$\frac{M_{t}}{M_{\text{∞}}}$ = 14.21t^0.45^	0.91
MBO-NE3	Korsmeyer–Peppas	$\frac{M_{t}}{M_{\text{∞}}}$ = 14.99t^0.43^	0.90

## Data Availability

The original contributions presented in the study are included in the article; further inquiries can be directed to the corresponding author.

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
