# Peer review of "Design, Optimization, Manufacture and Characterization of Milbemycin Oxime Nanoemulsions"

_pharmaceutics, 2025, doi:10.3390/pharmaceutics17030289_

Round 1
Reviewer 1 Report
Comments and Suggestions for Authors
The current research manuscript focusing on the development of nanoemulsion formulation for milbemycin oxime is interesting matches with the journal standards. However, the quality of the manuscript and presentation of the data is not satisfactory. Authors have to improve the quality of the manuscript by providing more detailed methodologies and by improving the English language and by correcting the manuscript for typos. Authors are requested to consider the below comments while revising the manuscript.
1. Authors are suggested to rewrite the background information more professionally rather than juts mentioning that no method is available for this drug compound. Please rewrite it such a way “the main focus of the current research is to develop a nanoemusion formulation……”
2. Line 16: Correct “prescription” to “preparation”
3. The entire manuscript requires English editing for grammatical mistakes and content flow issues.
4. What sought of medication is currently available in the market for milbemycin oxime? What are the disadvantages of the current formulation and why there’s a need for developing oil in water emulsion.
5. Section 2.2: was the method developed in-house? If not please provide the reference.
6. Section 2.4 and 2.5: The methods needs to be improved and authors are suggested not to discuss the results within the methodology section.
7. Authors are suggested to avoid using the term “prescription”
8. What was the pH of in vitro release medium?
9. Authors are suggested to include a section for stability studies within the methodologies.
Comments on the Quality of English LanguageThe quality of the manuscript is not satisfactory. The entire manuscript needs to be corrected for typos and revised for English grammatical mistakes and content flow.
Reviewer 2 Report
Comments and Suggestions for Authors
The authors developed an optimized nanoemulsion formulation containing milbemycin oxime, a macrolide antibiotic used in animals such as dogs. They characterized the physicochemical properties of the antibiotic nanoformulation using standard experimental procedures. The study objectives/theme were well addressed, and the following corrections are recommended:
· Abstract: A sentence or phrase justifying why developing a milbemycin oxime nanoemulsions is important should be added. Why milbemycin oxime and not another drug?
· Please include particle size measurements from the TEM analyses and compare this to the hydrodynamic sizing captured using the DLS method.
· From literature, a zeta potential of -30 or +30 or higher indicate nanoformulation stability so reporting values contained in the manuscript (i.e., -4 to -7 mV) mean formulation instability. Can authors add a justification of why they described their formulation as stable?
Overall, the manuscript was well written.
Round 2
Reviewer 1 Report
Comments and Suggestions for Authors
All the comments are well addressed with proper justification and supporting literature. The revised version can be accepted for publication.
Author Response
Thank you again for your positive comments on the revision of our manuscript.